# Metallomic Profile of Placental Tissue and Its Association with Maternal and Neonatal Parameters: A Cross-Sectional ICP-OES Study in Lower Silesia

**DOI:** 10.3390/ijms262411985

**Published:** 2025-12-12

**Authors:** Aleksandra Kuzan, Emilia Królewicz, Marta Kardach, Justyna Rewak-Soroczyńska, Małgorzata Kowalska, Aldona Molęda, Rafal J. Wiglusz

**Affiliations:** 1Faculty of Medicine, Wroclaw University of Science and Technology, ul. Hoene-Wrońskiego 13 c, 58-376 Wrocław, Poland; 2Department of Medical Biochemistry, Wroclaw Medical University, Chalubinskiego 10, 50-368 Wroclaw, Poland; emilia.krolewicz@umw.edu.pl; 3Institute of Low Temperature and Structure Research, Polish Academy of Sciences, Okolna 2, 50-422 Wroclaw, Poland; 4Faculty of Applied Studies, DSW University of Lower Silesia, Strzegomska 55, 53-611 Wroclaw, Poland; 5I Department of Gynecology and Obstetrics, Wroclaw Medical University, Borowska 213, 50-368 Wroclaw, Poland

**Keywords:** trace elements, placenta, macroelements, microelements, metallomics

## Abstract

The placenta, a temporary organ that connects mother and child for nutrient and metabolite exchange, becomes medical waste after birth but can provide valuable metabolic insights. Thirty-three placenta samples were analyzed using ICP-OES to determine concentrations of ten elements, including macro-, micro-, trace, and heavy metals. Results were compared with maternal and neonatal data, including Apgar scores, maternal age, and blood parameters. Correlations were found between elements (e.g., Ca–Mg, Fe–Zn, and Mn–Cu) and between mineral levels and maternal or infant parameters (e.g., Ca–RBC, Mn–Hb, Cu–PLT, and Cu–UA Pi). No quantifiable heavy metals were detected, nor associations with smoking, gestational diabetes, preterm birth, birth weight, or Apgar scores. Findings suggest that maintaining proper blood morphology and preventing anemia in pregnancy requires attention not only to iron but also to Ca^2+^, Mg^2+^, and Mn^2+^ levels. Manganese and copper assessment may be beneficial for diagnostic purposes in pregnant women. Further large-scale tissue studies are recommended, including comprehensive maternal–fetal health data such as Doppler velocimetry of placental vessels.

## 1. Introduction

Proper hygiene in the mother’s lifestyle affects the balanced growth and development of the fetus. The placenta is a highly metabolically active organ and serves as an interface between maternal and fetal circulation. It is evident that the deficiency of key macro- and micronutrients in the placenta may have very serious health consequences for the baby, ranging from various developmental defects to death.

Calcium and magnesium are necessary for the mineralization of the bones of the fetus [1]. Calcium is also responsible for blood clotting and transmitting nerve signals, and magnesium regulates the activity of many enzymes and the contractility of the uterine muscles [2,3]. Among nutritional mistakes of pregnant women, it is worth noting the insufficient supply of calcium and magnesium from the diet, which may lead to the development of hypertension and pre-eclampsia [2,3]. It is often the case that calcium homeostasis in tissues is not achieved, leading to premature calcification of the placenta, which in turn impairs blood circulation in the tissue and the supply of oxygen and nutrients to the child [4,5]. Therefore, the research was focused on assessing whether placental calcium levels, particularly when subclinically increased or decreased, are associated with poorer newborn parameters. Similarly, in the case of magnesium, we investigate what dependencies determine or result in the accumulation or deficiency of this element in the placenta.

Microelements are also important components of the placenta: copper, iron, manganese, zinc, chromium, cobalt, and nickel were selected. Their common feature is that the body’s need for them is normally very low—generally below 100 mg/day [6]. Although their functions are different, virtually each of them is a cofactor of various enzymes. In fetal development, iron is particularly important because it enables gas exchange between the mother and the fetus. It is reported that the placenta has the capacity to store iron in resident reticuloendothelial cells to provide a buffer against periods of low maternal iron supply [7]. Anemia may arise not only as a result of iron deficiency, but also copper deficiency, since the Cu^2+^ plays an important role in the absorption and metabolism of Fe^2+^ and Fe^3+^ [5]. Although cobalt is an ultra-trace element, as a component of vitamin B_12_ (cobalamin), it appears to be crucial for the well-being of both mother and child. Co^2+^ is responsible for myelinogenesis and erythrocyte production, thus its deficiency leads to neurological damage and anemia [8]. A large proportion of the above-mentioned metals play a key role in mechanisms related to defense against free radicals and the development of inflammation, especially manganese (II) and zinc (II) ions [9]. Chromium (III) ion is involved in the metabolism of carbohydrates; thus, its deficiency is associated with the occurrence of gestational diabetes mellitus, and its excess is associated with low birth weight and other fetal defects [5]. A detailed description of the function of each micronutrient in the placenta was presented in review articles, such as the work of Khayat et al. (2017), Hofstee et al. (2018) or Farias et al. (2020) [2,9,10].

For this project, in addition to macro- and micronutrients, two examples of heavy metals that are not only unnecessary for the functioning of the body but are also harmful were selected: lead and cadmium. Some micronutrient metals, accumulated in excess, may have a similar effect, e.g., nickel or chromium [11]. It is reported that such metals are found in a variety of tissues due to adsorption from polluted air, food, or water [11,12,13,14,15]. There are also studies in the context of the placenta, where heavy metal accumulation in the tissue has been found to be associated with a higher risk of miscarriage, as well as risk of congenital malformations and reduced birth weight [5,11,16]. Interestingly, some studies have shown that the levels of some of these metals, such as cadmium, were not higher in placentas of women with complications such as gestational diabetes; on the contrary, they were lower than in the control group [15]. Therefore, it is considered that these results were particularly controversial and worth repeating in a different group of patients. This work analyzed whether exposure to lead, cadmium, and nickel constituted a real threat to the life and health of newborns in the studied group of pregnant patients living in Lower Silesia (Poland). The aim of this project was to investigate the relationship between the content of the selected elements and the birth parameters related to the health of mothers and newborns (Apgar score, body weight, body length, head and chest circumference, neonatal diagnosis). Eleven elements were selected: calcium (Ca^2+^), magnesium (Mg^2+^), iron (Fe^3+^), copper (Cu^2+^), chromium (Cr^3+^), manganese (Mn^2+^), cadmium (Cd^2+^), lead (Pb^2+^), zinc (Zn^2+^), cobalt (Co^2+^), and nickel (Ni^2+^). Macro-, microelements, and heavy metals were analyzed in the placental samples of women in various physiological states and with different courses of pregnancy, to be able to draw conclusions about the consequences of disturbances in the elemental composition of the placenta.

## 2. Results

Four elements did not exceed the LOD (limit of detection) in any of the samples: cadmium (II) (LOD < 0.13 µg/L), cobalt (II) (LOD < 0.38 µg/L), nickel (II) (LOD < 0.69 µg/L), and lead (II) (LOD < 2.4 µg/L) ions. Therefore, the hypothesis concerning the influence of cigarette smoking on the accumulation of heavy metals in the placenta remained unconfirmed.

The analysis of relationships between individual analytes in the samples of placenta allowed the identification of the following statistically significant positive correlations: very high correlations for Ca-Mg, Fe-Mn, and Mg-Zn pairs; high correlations for pairs Ca-Zn, Cr-Fe, Fe-Zn, and Mn-Zn; moderately strong correlations for Cr-Zn, Cu-Fe, and Cu-Mn pairs. No statistically significant negative correlations were observed. The correlation coefficients for each pair of elements are presented in Table 1.

Contrary to expectations, no statistically significant correlations were observed between the newborn’s weight or the Apgar scale and the content of the analyzed elements in the placenta. An illustration of the relationship between the weight of a newborn and the content of the analyzed analytes is shown in Figure 1.

Similarly, no statistically significant correlations were observed between the length of hospitalization, maternal BMI, or maternal age and the content of the analyzed elements in the placenta. Figure 2 illustrates the relationship between maternal age and the content of the analyzed analytes. Analogous graphs are obtained for the above-mentioned relationships.

Contrary to expectations, no relationship was observed between iron content and prepartum RBC, HGB, and MCV. However, positive correlations were observed between Ca^2+^ and RBC, Mg^2+^ and RBC, Mn^2+^ and HGB, Mn^2+^ and HCT, and three pairs of negative correlations: between Cu^2+^ and UA Pi, Cu and PLT, and between Fe^3+^ and PLT. Detailed results for the correlation between the parameters discussed are presented in Table 2.

A graphical representation of the interdependence of three parameters: RBC, Ca^2+^, and Mg^2+^ is presented as a scatter plot in Figure 3. Most of the samples concern cases of deliveries between 38 and 40 weeks of gestation, and most of them are characterized by low levels of calcium (II) and magnesium (II) ions. It can be identified as individual outliers: one patient has significantly elevated calcium levels in the placenta, probably due to advanced tissue calcification. Similarly, only one patient delivered prior to 30 weeks of gestation. Some deviations from the mean can also be observed in the case of three patients in the context of the magnesium content in the sample. Moreover, it can be noted that there were no outliers removed for statistical analysis. Summarizing, with a larger study group, the distribution of results could be more extended between the axes. A similar graph for the set of variables: maternal RBC, Ca^2+^, and Mg^2+^ in the placenta is shown in Figure 4. Patients with a high level of Ca^2+^ are also high in Mg^2+^, and inversely, as previously has been shown by the Spearman correlation.

Testing the hypotheses concerning the influence of cigarette smoking on the content of metals in the placenta, the influence of the content of the analyzed elements in the placenta and the occurrence of premature birth, and the correlation between the occurrence of gestational diabetes and the content of metals in the placenta led to the conclusion that no statistically significant relationship was found. The results of this part of the analysis are presented in Table 3.

## 3. Discussion

At the beginning, reference will be made to the results concerning heavy metals. It was expected to find their accumulation in samples where pregnancy complications were observed, as they are considered metals that induce oxidative stress [15]. Surprisingly, no cases of cadmium, cobalt, nickel, or lead were observed in any of the placental samples. The area from which the patients come is moderately urbanized, but it is not free of exhaust fumes or heavy industry. Based on previous experiments with other types of tissues collected from inhabitants of the same area (Lower Silesia, Poland), the presence of at least cadmium and lead in at least some samples was expected. It turns out that these elements are accumulated in some arterial samples, and they were not detected in the analyzed placentas, even though the determination methodology and equipment used for both projects were the same [17]. Other publications also report on the presence of heavy metals in the placenta. Among them, the publication by Mazurek et al. is especially valuable for comparison, because it is based on analysis of placentas taken from women in the same hospital; accumulation of cadmium and lead was observed there; cadmium, especially in smoking women [4]. The interpretation of the result can be based on the assumption that (1) the environment of the women who underwent the study was not contaminated with heavy metals; (2) the study group was too small to be considered representative, especially the subgroup of female smokers. It is recognized that the main limitation of this project is the relatively small study sample, and the project itself is treated rather as a pilot study, with potential only for drawing preliminary conclusions.

Part of the relationships between individual metals was observed both in the placenta in the project and in other tissues analyzed in previous projects. Correlation of the Fe–Zn pair was observed, as in placentas, also in arteries (in both sexes). Moreover, correlations between Mn–Zn and Cu–Mn pairs were also observed in male arteries, and a correlation of the Cu–Fe pair was also observed in female arteries [17]. On the other hand, in the thyroid glands, as in the placentas, a correlation of the Ca–Mg pair was observed too [18]. Some of these correlations are also reported in other tissues, for example, in blood serum [19,20], the umbilical cord, and the fetal membrane [21]. Especially noteworthy is the dependence of the Ca-Mg pair, which obtained a very high correlation (r close to 0.9). At first glance, it may be surprising that these two elements correlate very strongly with each other, even though Mg^2+^ and Ca^2+^ ions are antagonists [21]. It is known that both calcium (II) and magnesium (II) ions are key building blocks of the fetal skeleton, and the demand for these elements in pregnant women increases significantly. A pathological situation, however, is the calcification of the placenta, i.e., the deposition of calcium (II) salts in the placenta [22]. On the other hand, hypermagnesemia also has toxic effects on pregnancy, leading to diminished deep tendon reflexes, apnoea, and electromechanical dissociation. High Mg^2+^ concentrations in the placenta may reduce Ca^2+^ transport to the fetus [21]. This pair of Ca–Mg elements must be in balance for the course of pregnancy to be physiological, and this is indeed usually the case, as the research showed.

The stated relationship between Fe^3+^ and Zn^2+^ can be considered in the context of interdependent pro- and antioxidant mechanisms occurring in placental tissues. Oxidative stress, i.e., shifting the balance towards excessive biological oxidation, can cause placental damage and failure, and consequently a premature birth [23]. Iron is of great importance here because Fe^2+/3+^ are involved in various reactions where free radicals can be formed as a by-product, e.g., electron transfer reactions during respiration in mitochondria, metabolism of xenobiotics; additionally, Fe^2+^ directly catalyzes the formation of reactive radicals via Fenton chemistry. It is therefore concluded that iron excess is believed to generate oxidative stress [24]. In turn, zinc participates in rather anti-oxidative mechanisms, mainly as a cofactor of superoxide dismutase (Zn-SOD) [25]. Similarly, Cu—this element also participates in scavenging free radicals as a component of SOD [25] and as a component of ferroxidases such as hephaestin (in the gut), ceruloplasmin (in the liver), and zyklopen (in the placenta)—enzymes involved in the oxidation of Fe^2+^ to Fe^3+^, which are stored in the cell or transported to the blood [26]. Therefore, it could be concluded that the positive correlation of Fe–Zn and Fe–Cu pairs was observed, indicating these elements usually remain in balance, allowing homeostasis to be achieved in the placenta. An interesting research problem would be to verify whether such a correlation is also observed in cases of pathological pregnancies.

An important problem in pregnant women is the difficulty in maintaining the right amount of iron and erythrocytes in the blood, and hence the frequent occurrence anemia—statistics show that about 1/3 of pregnant women suffer from anemia in the third trimester [27]. In this study, a significant relationship was observed between blood count parameters and the level of certain elements in the placenta: the amount of manganese correlates with the maternal hemoglobin concentration and hematocrit, and the number of maternal erythrocytes before delivery positively correlates with the content of Ca^2+^ and Mg^2+^. Interestingly, none of the parameters related to the diagnosis of anemia show a relationship with the level of iron, which is the most obvious component of erythrocytes. The regulation of the amount of Fe^3+^ in the placenta is constrained by specific control mechanisms that allow the functioning of the placenta and the embryo. Coming back to the relationship between Mn–Hb and Mn–HCT, manganese is a microelement that performs a certain function in the metabolism of red blood cells, not least because of its chemical similarity to iron (III) ions and the resulting consequences regarding the competition of these two elements as part of absorption from the intestine [28]. On the other hand, erythrocytes are responsible for Mn^2+^ distribution due to their ability to carry the Mn^2+^ with the presence of various Mn^2+^ transporters [29]. Blood Mn^2+^ level has been reported to be higher in patients with anemia or iron deficiency; however, some reports, e.g., Kim et al. and Liu et al., showed, as in the case of this work, a positive correlation between Mn- and Hb, at least in patients with chronic kidney disease [30,31]. The context of pregnancy seems not to be discussed in the literature; thus, the data describing the relationship between the placental Mn content and the amount of hemoglobin and hematocrit in the mother’s blood are available for the scientific community’s verification.

For erythrocytes, the correlations demonstrated between RBC–Ca and RBC–Mg should also be discussed. It is well known that Mg^2+^ is important for the hematopoietic system. Consequently, low levels of magnesium are linked to a higher risk of anemia in pregnant women, and magnesium consumption was inversely associated with anemia [32]. The calcium (II) ion also participates in hematopoiesis, regulating the cell cycle, structural integrity, motility, and volume of erythroblasts and erythrocytes [33]. The result of positive correlations between the Ca^2+^ and Mg^2+^ content in the placenta and the RBC of the mother obtained by us can therefore be considered logical, although surprisingly not previously reported in other scientific studies.

In the literature, there are also no reports of a negative correlation between thrombocytes in maternal blood and Cu and Fe in the placenta; this is probably the first time it has been reported. Although the PLT–Fe relationship is not obvious and direct, it is indeed found that thrombocytopenia may occur in iron deficiency anemia, which may explain our result [34]. The PLT–Cu relationship is also described as inversely proportional. Copper influences morphology in some patients with anemia, at least sickle cell anemia, and possibly in other types of anemia as well, likely by inducing red cell haemolysis, oxidant tissue damage, and stimulating the immune system [35].

A very interesting result is the negative Cu–UA Pi correlation. It is worth noting that abnormal Doppler velocimetry of the UA is a particularly valuable diagnostic tool, as UA Pi increases with the vascular disease of the placenta [34]. According to the data, the less copper in the bearing, the less favorable this parameter is. Few studies have focused on the correlation between the placental chemical element concentrations and Doppler markers of placental function; in fact, only one paper was found, by Gómez-Roig et al. [36]. This research significantly differs in the study group—its size, maternal residence, and thus exposure to heavy metals, and the profile of fetal patients. The results are consistent in that lower copper levels are associated with abnormal fetal–maternal Doppler results [36].

When discussing the results in terms of gestational diabetes mellitus (GDM), it is worth comparing the results with the research of the Roverso team, which focused on placental research in this context [1,15]. Although differences between some elements, e.g., chromium, were expected in control samples and samples from people with diabetes, no significant differences were found. Both the work by Roverso and the present study indicate that placental metal content—including chromium—is not a reliable, repeatable marker for gestational diabetes (GDM), as the element concentrations do not differ significantly between healthy pregnancies and GDM cases. Earlier observations (Cd and Se) [15] have not been confirmed in subsequent studies [1]. Roverso and colleagues reported that it is umbilical cord blood rather than the placenta itself that is the material in which differences are observed. In whole cord blood of GDM cases, elements including Ca, Cu, Na, and Zn were found at higher concentrations than in controls, while Fe, K, Mn, P, Rb, S, and Si showed lower concentrations [1].

## 4. Materials and Methods

### 4.1. Study Material and Ethics Statement

A total of 33 samples of maternal placentas were obtained from 33 pregnant woman who gave birth in the obstetric–gynecological ward of the First Department of Gynecology and Obstetrics of the Wroclaw Medical University in 2015–2016. The only inclusion criteria were the delivery of a live pregnancy between 22 and 42 weeks of gestation. The exclusion criteria were subjects with multiple pregnancies. Sociodemographic and other data on mothers and children were obtained from research project questionnaires, medical histories, and hospital discharge summaries for mothers and children. The patient’s history provided data concerning age, gestational age, body weight, body height, birth method, diagnosis, blood parameters, the course of pregnancy, and drugs used during the pregnancy. Among the diseases complicating pregnancy in women, the most common were diagnosed gestational diabetes mellitus GDM 1 and GDM 2, diabetes G1 and G2, hypothyroidism, nicotinism, and preterm premature rupture of membrane (PROM). The parameters measured in newborns included gender, birth weight, body length, head circumference, chest circumference, Apgar score, and neonatal diagnosis. The parameters regarding patients and newborns are presented in Table 4 and Table 5.

### 4.2. Analytical Procedure Using Inductively Coupled Plasma Optical Emission Spectrometry (ICP-OES)

Determination of trace elements such as calcium, magnesium, zinc, copper, iron, manganese, and chromium, as well as toxic metals like cadmium and lead, was carried out using inductively coupled plasma optical emission spectrometry (ICP-OES). The total concentration of the trace elements was measured using an Agilent 5110 synchronous vertical dual view (SVDV) ICP-OES instrument (Agilent Technologies, Inc., Santa Clara, CA, USA) equipped with an easy-fit quartz torch with standard 1.8 mm injector and a Seaspray nebulizer as a sample introduction system and a double-pass glass cyclonic spray chamber. The samples were subjected to the procedure described in Kuzan et al. (2021) [17] with the following modification: they were washed three times in de-ionized water (to remove blood and standardize the condition of tissues for analysis, because each of them was bled to a different extent), air dried for two hours, and weighed. Next, the key step was to digest the sample. The samples were transferred to a Teflon vessel containing 25 mL 1 M ultrapure HNO_3_ (Sigma-Aldrich, Saint Louis, MI, USA) and placed in a microwave reactor for 90 min. In microwave-stimulated hydrothermal conditions, under autogenous pressure of 25 atm and at 250 °C, the procedure was repeated for every specimen. The concentrations of elements were expressed as mg/L, and then recalculated to mg/g of tissue.

### 4.3. Statistical Analysis

The relationships between individual features and analytes were analyzed using Kendall’s Tau-b correlation coefficient (for the length of hospitalization and for the Apgar scale) or Spearman’s (for the rest of the parameters), with outliers for the analyzed microelements not removed. To examine the relationship between the content of micronutrients and the occurrence of binary features (diabetes, smoking, and premature birth) and the impact on the risk of occurrence of 0/1 features, the point-biserial correlation coefficient and the logistic regression model were used.

Statistica software system (version 13.3, StatSoft, TIBCO Software Inc., Palo Alto, CA, USA) was used for the analysis. The significance level was set at *p* > 0.05.

## 5. Conclusions

This study successfully characterized the profile and interrelationships of essential trace elements (calcium, magnesium, zinc, copper, iron, manganese, and chromium) in the placental tissue of pregnant patients residing in Lower Silesia, Poland. A highly significant positive correlation was confirmed between placental Ca^2+^ and Mg^2+^ concentrations, emphasizing the tight homeostatic regulation of these critical elements during pregnancy. Significant inter-elemental relationships were also established for other pairs (e.g., Cu^2+^ and Fe^3+^), highlighting the complexity of metal transport and interaction within the placental barrier. The findings suggest that the placental concentrations of these essential elements serve as an important biomarker of fetal exposure and nutritional status. Even moderate deviations from optimal levels—which may not be clinically apparent in the mother—can influence the parameters of the newborn.

These results underscore the need for further longitudinal studies to establish reference ranges for placental metallomics and to confirm whether elemental dysregulation is a causative factor or a consequence of underlying placental dysfunction affecting fetal growth and development.

## Figures and Tables

**Figure 1 ijms-26-11985-f001:**
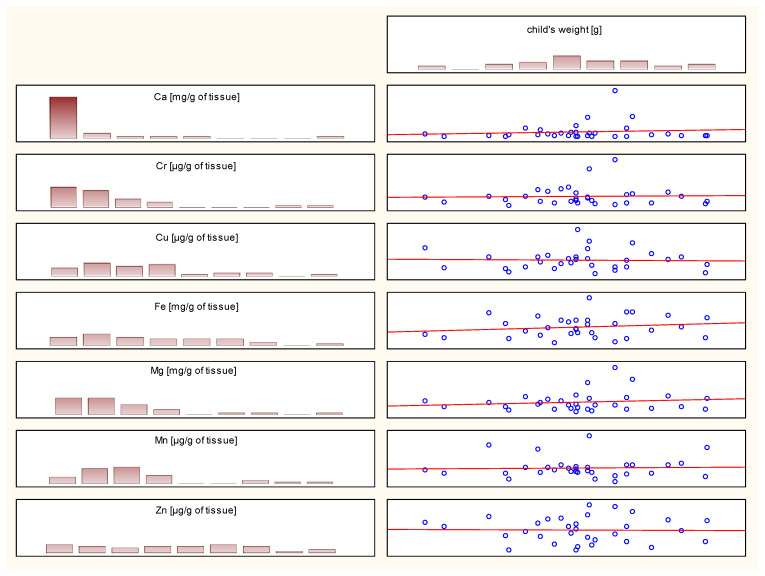
Correlation plot depending on the weight of the newborn (Spearman’s correlation).

**Figure 2 ijms-26-11985-f002:**
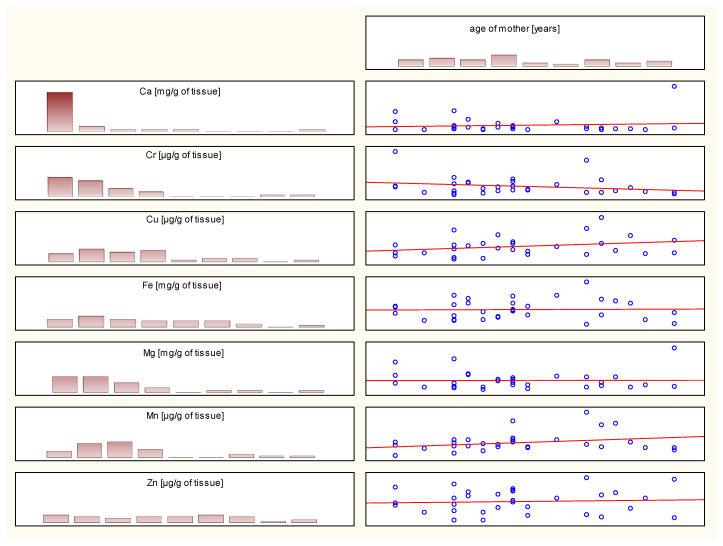
Correlation plot depending on the age of the mother (Spearman correlation).

**Figure 3 ijms-26-11985-f003:**
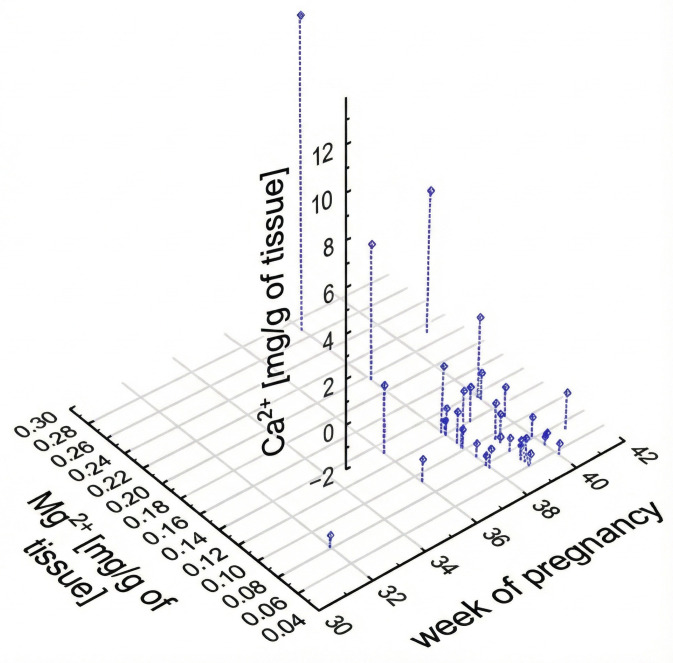
Scatterplot of parameters: Ca^2+^ and Mg^2+^ in the placenta and length of pregnancy.

**Figure 4 ijms-26-11985-f004:**
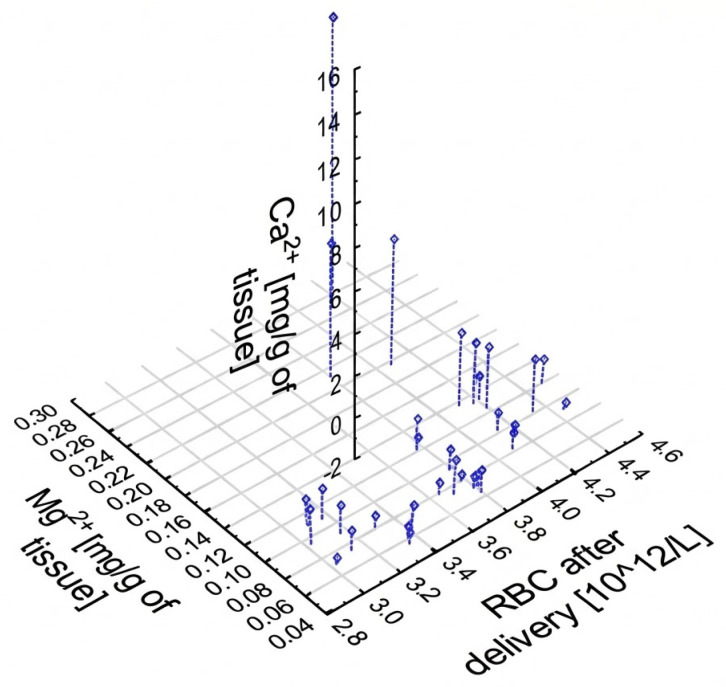
Scatterplot of parameters: maternal RBC and placental Mg^2+^ and Ca^2+^.

**Table 1 ijms-26-11985-t001:** Spearman’s correlation matrix between the contents of the analyzed metals. The table contains the parameter r and *p*; results of r for which *p* > 0.05 are marked in red.

Ions [mg/g]	Ca^2+^ [mg/g]	Cr^3+^ [µg/g]	Cu^2+^ [µg/g]	Fe^3+^ [mg/g]	Mg^2+^ [mg/g]	Mn^2+^ [µg/g]	Zn^2+^ [µg/g]
Ca^2+^ [mg/g]	1.0000	−0.1189	0.0004	−0.0667	0.9097	−0.107	0.5305
	*p* = ---	*p* = 0.517	*p* = 0.998	*p* = 0.717	* p * = 0.000	*p* = 0.559	* p * = 0.002
Cr^3+^ [µg/g]	−0.1189	1.0000	0.1778	0.6290	0.0799	0.3234	0.4119
	*p* = 0.517	*p* = ---	*p* = 0.330	* p * = 0.000	*p* = 0.664	*p* = 0.071	* p * = 0.019
Cu^2+^ [µg/g]	0.0004	0.1778	1.0000	0.4453	0.1319	0.4074	0.2323
	*p* = 0.998	*p* = 0.330	*p* = ---	* p * = 0.011	*p* = 0.472	* p * = 0.021	*p* = 0.201
Fe^3+^ [mg/g]	−0.0667	0.6290	0.4453	1.0000	0.2125	0.7463	0.5816
	*p* = 0.717	* p * = 0.000	* p * = 0.011	*p* = ---	*p* = 0.243	* p * = 0.000	* p * = 0.000
Mg^2+^ [mg/g]	0.9097	0.0799	0.1319	0.2125	1.0000	0.1101	0.7586
	* p * = 0.000	*p* = 0.664	*p* = 0.472	*p* = 0.243	*p* = ---	*p* = 0.549	* p * = 0.000
Mn^2+^ [µg/g]	−0.1072	0.3234	0.4074	0.7463	0.1101	1.0000	0.5581
	*p* = 0.559	*p* = 0.071	* p * = 0.021	* p * = 0.000	*p* = 0.549	*p* = ---	* p * = 0.001
Zn^2+^ [µg/g]	0.5305	0.4119	0.2323	0.5816	0.7586	0.5581	1.0000
	* p * = 0.002	* p * = 0.019	*p* = 0.201	* p * = 0.000	* p * = 0.000	* p * = 0.001	*p* = ---

**Table 2 ijms-26-11985-t002:** Spearman’s correlation matrix between selected analyzed biochemical parameters and contents of analyzed metals. Parameter r is given; results for which *p* > 0.05 are marked in red.

	Mean	SD	Ca^2+^ [mg/g of Tissue]	Cr^3+^ [µg/g of Tissue]	Cu^2+^ [µg/g of Tissue]	Fe^3+^ [mg/g of Tissue]	Mg^2+^ [mg/g of Tissue]	Mn^2+^ [µg/g of Tissue]	Zn^2+^ [µg/g of Tissue]
umbilical artery pulsatility index (UA Pi)	0.81	0.20	−0.14	0.07	−0.57	−0.25	−0.12	−0.03	−0.10
middle cerebral artery pulsatility index (MCA-Pi)	1.37	0.32	−0.52	−0.24	−0.52	−0.29	−0.46	0.20	−0.33
white blood cells (WBC) ^a^ [10^9^/L]	10.19	2.22	0.44	0.20	−0.20	−0.12	0.50	−0.04	0.33
red blood cells (RBC) ^a^ [10^12^/L]	4.06	0.27	0.61	−0.15	−0.25	0.10	0.57	0.05	0.16
hemoglobin (HGB) ^a^ [g/dL]	11.96	0.74	0.19	0.23	0.08	0.37	0.40	0.55	0.46
hematocrit (HCT) ^a^ [%]	35.34	1.99	0.30	0.29	0.24	0.49	0.47	0.57	0.49
platelets (PLT) ^a^ [10^9^/L]	184.99	68.90	0.00	−0.10	−0.60	−0.56	−0.06	−0.31	−0.14
mean corpuscular volume (MCV) ^a^ [fl]	95.09	32.79	−0.23	0.06	0.37	0.18	−0.26	0.03	−0.12

^a^ measurement before delivery.

**Table 3 ijms-26-11985-t003:** Point-two-series correlations for smoking, diabetes, and premature birth in the context of the selected metals in the placenta.

Ions	Diabetes 0/1	Nicotine Dependence 0/1	Premature Delivery 0/1
Ca^2+^ [mg/g of tissue]	−0.1835	−0.0162	−0.0498
	*p* = 0.315	*p* = 0.930	*p* = 0.786
Cr^3+^ [µg/g of tissue]	0.2032	−0.1241	0.0050
	*p* = 0.265	*p* = 0.499	*p* = 0.978
Cu^2+^ [µg/g of tissue]	−0.0770	−0.0765	−0.1399
	*p* = 0.675	*p* = 0.677	*p* = 0.445
Fe^3+^ [mg/g of tissue]	0.1275	−0.2491	−0.0500
	*p* = 0.487	*p* = 0.169	*p* = 0.786
Mg^2+^ [mg/g of tissue]	−0.1272	−0.0263	−0.0249
	*p* = 0.488	*p* = 0.886	*p* = 0.893
Mn^2+^ [µg/g of tissue]	−0.0752	−0.1466	0.1439
	*p* = 0.683	*p* = 0.423	*p* = 0.432
Zn^2+^ [µg/g of tissue]	−0.0266	−0.1251	0.0707
	*p* = 0.885	*p* = 0.495	*p* = 0.701

**Table 4 ijms-26-11985-t004:** Information on pregnant women (*n* = 33).

Variable	Pregnant Women (*n* = 33)
Age (years)	29.5 ± 5.26 ^a^
Body weight (kg)	81 ± 16.78 ^a^
Body height (cm)	165.8 ± 6.32 ^a^
BMI (kg/m^2^)	29.4 ± 5.52 ^a^
Parity	
Nulliparous	13 (39.4%) ^b^
Multiparous	20 (60.6) ^b^
Gestational age (weeks)	38.6 ± 1.63 ^a^
Birth method	
Spontaneous	2 (6.1%) ^b^
Caesarian section	31 (93.9%) ^b^
Comorbidities	
Diabetes	10 (30.3%) ^b^
G1	2 (6.1%) ^b^
GDM 1	4 (12.2%) ^b^
G2	2 (6.1%) ^b^
GDM 2	2 (6.1%) ^b^
Hypothyroidism	7 (21.2%) ^b^
Nicotinism	3 (9.1%) ^b^
PROM	4 (12.2%) ^b^

^a^ Data were presented as mean ± standard deviation. ^b^ Data were presented as *n* (%).

**Table 5 ijms-26-11985-t005:** Information on newborn (*n* = 33).

Variable	Newborn (*n* = 33)
Gender	
Female	14 (42.4%) ^a^
Male	19 (57.6%) ^a^
Birth weight (g)	3299.85 ± 503.65 ^b^
Body length (cm)	54.03 ± 3.30 ^b^
Head circumference (cm)	33.15 ± 1.44 ^b^
Chest circumference (cm)	33.21 ± 2.0 ^b^
Apgar score	9.76 ± 0.67 ^b^
Neonatal diagnosis	
Prematurity	2 (6.1%) ^a^
Healthy newborn	22 (66.7%) ^a^
Low birth weight	2 (6.1%) ^a^
Increased jaundice	3 (9.1%) ^a^
Asphyxia	2 (6.1%) ^a^
Inborn pneumonia	2 (6.1%) ^a^
Other complications	4 (12.2%) ^a^

^a^ Data were presented as *n* (%). ^b^ Data were presented as mean ± standard deviation.

## Data Availability

The raw data supporting the conclusions of this article will be made available by the authors on request.

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
