# Peer review of "Metallomic Profile of Placental Tissue and Its Association with Maternal and Neonatal Parameters: A Cross-Sectional ICP-OES Study in Lower Silesia"

_ijms, 2025, doi:10.3390/ijms262411985_

Round 1
Reviewer 1 Report
Comments and Suggestions for Authors
In present work, Kuzanet al. try to investigates the elemental composition of placental tissues and explores potential correlations with maternal and neonatal health parameters. However, the study is limited by its small sample size and many low writing wrongs, as well as the novelty. In addition, there are some questions that should be answered.
Major concerns
- There are some publications very related with this manuscript are not included in order to demonstrate the novelty of this manuscript. Therefore, the novelty of this manuscript is doubtful. For example,
Roverso M, Berté C, Di Marco V, Lapolla A, Badocco D, Pastore P, Visentin S, Cosmi E. The metallome of the human placenta in gestational diabetes mellitus. Metallomics. 2015;7(7):1146-54.
Harrington JM, Young DJ, Fry RC, Weber FX, Sumner SS, Levine KE. Validation of a Metallomics Analysis of Placenta Tissue by Inductively-Coupled Plasma Mass Spectrometry. Biol Trace Elem Res. 2016;169(2):164-73.
Roverso M, Di Marco V, Badocco D, Pastore P, Calanducci M, Cosmi E, Visentin S. Maternal, placental and cordonal metallomic profiles in gestational diabetes mellitus. Metallomics. 2019;11(3):676-685.
- The small sample size (n=33) is limited, especially for subgroup analyses (smokers, diabetics).
- Line 340, the method section mentions washing the sample three times with deionized water to remove blood. What is the specific time for cleaning? How to determine if cleaning has been "standardized"? Has the cleaning effect been validated (such as measuring hemoglobin or iron content in the cleaning solution)?
- The ethical approval number mentioned in the statement is KB-418/2022, but the research samples were collected between 2015-2016. Please explain why the ethical approval for 2022 can retroactively approve research from 2015-2016?
- English writing should be checked and revised throughout the manuscript. There are many low wrongs. Some sentences are long or awkwardly phrased. Consider professional editing for clarity and flow. There are many repeat references.
Minor concerns
- Introduction section is too long, which should be refined. There are ‘The aim of’ (Line 45), ‘our goal is’ (Line 64), and ‘Our goal was’ (Line 96).
- Line 70, correct ‘100 mg / day’.
- Line 86, correct ‘(2017) or Farias’ to ‘(2017) and Farias’.
- Line 110, please explain ‘LOD’.
- Lines 122, correct ‘p>0.05’, ‘P’ should be italic. Please check and revise it throughout the manuscript.
- Correct Figure 1 or Fig. 2. In addition, the words in the images need to be moved down a bit. Furthermore, the figure legends are needed for all figures.
- Table 2, there is an issue with the layout.
- Line 343, correct ‘HNO3’.
- Lines 312-314, 391-393, there is a repeat.
- Conclusions should be refined. In general, it should not include reference.
- Reference section. Repeat references and inconsistent reference format.
For example, Ref. 1, ‘J Clin Diagn Res’.
Ref. 3, ‘Journal of Trace Elements in Medicine and Biology’.
Ref. 19 and 20 are one paper.
Ref. 22 and 25 are one paper.
Ref. 38 and 39 are one paper.
Comments on the Quality of English LanguageThe English could be improved to more clearly express the research.
Reviewer 2 Report
Comments and Suggestions for Authors
Review of the manuscript titled "Metallomics in placentae" (ID: ijms-3976490). My suggestions for the manuscript:
- In my opinion, the abstract is sufficient.
- The introduction is also sufficient – the authors have justified why they determined specific micro-, macro-, and heavy metals in their study. However, I believe the aim of the work should be placed at the end of the introduction, as this will encourage the reader to read the manuscript further. However, I leave this decision to the authors. However, I believe the work should have a single, clearly stated aim, and the aim of the study is stated several times in the introduction. I believe this should be improved.
- Lines 309-312: This text contains information about the source of the material twice. I believe that once is sufficient.
- In Table 4, the authors should eliminate column 2. Instead, they should include a superscript reference for each variable, e.g., A, B, and explain below the table that reference A: data were presented as mean ± standard deviation, B: n and %. The table is also missing a percentage value, which should be added. Do the rank values in this table contribute anything to this study? I suggest considering removing them. The same applies to Table 5.
- Please provide more details on the validation of the assays used to measure trace elements. Please include any quality control or reproducibility metrics, if available.
- In the methodology, please specify the method for obtaining sociodemographic and other data on mothers and children. The procedure should be described in detail.
- Lines 110-112. LOD values should be included in the methodology for all elements.
- Table 1: What were the p values?
- Table 2 also requires corrections. It's unreadable. Why write "before delivery" everywhere when it can be defined in the methodology?
- Generally speaking, the presentation of results needs improvement to be clear and legible for the reader.
- Why do the authors cite unpublished literature? In such a rich literature on the study's topic, I believe it's inappropriate to cite studies that haven't been peer-reviewed and published.
- In my opinion, the discussion is chaotic and disorganized. This applies to the entire text. Considering the rigor of scientific research, the entire manuscript requires thorough reformatting to provide a step-by-step introduction to the study and its rationale (introduction), present the research process (methodology), describe the obtained results (results), and discuss the authors' own results with those of other authors (discussion).
- The conclusions are too broad. The conclusion is the answer to the research question.
To sum up, due to the shortcomings mentioned above and the lack of a rigorous approach to a scientific study such as a scientific article, I propose not to accept the manuscript for publication in the journal in its current form.
Round 2
Reviewer 1 Report
Comments and Suggestions for Authors
Thanks for author’s responses. However, there are STILL some questions that should be answered, and the authors do not false this Reviewer.
- At the first review comments. This Reviewer suggests correcting ‘Repeat references and inconsistent reference format’.
However, the authors do not correct ‘Repeat references and inconsistent reference format’.
For example, Ref. 4, ‘Journal of Trace Elements in Medicine and Biology’.
Ref. 27 and 28 are one paper.
- At the first review comments. This Reviewer suggests that ‘There are some publications very related with this manuscript are not included in order to demonstrate the novelty of this manuscript.’
However, the authors only respond for ‘The recommended literature references have been supplemented in the manuscript.’
This Reviewer doubts the novelty of the article, because many similar researches related to ‘Metallomics in placentas’ had been published.
Roverso M, Berté C, Di Marco V, Lapolla A, Badocco D, Pastore P, Visentin S, Cosmi E. The metallome of the human placenta in gestational diabetes mellitus. Metallomics. 2015;7(7):1146-54.
Harrington JM, Young DJ, Fry RC, Weber FX, Sumner SS, Levine KE. Validation of a Metallomics Analysis of Placenta Tissue by Inductively-Coupled Plasma Mass Spectrometry. Biol Trace Elem Res. 2016;169(2):164-73.
Roverso M, Di Marco V, Badocco D, Pastore P, Calanducci M, Cosmi E, Visentin S. Maternal, placental and cordonal metallomic profiles in gestational diabetes mellitus. Metallomics. 2019;11(3):676-685.
- There are STILL many low wrongs. For example,
Fig. 3. Scatterplot of parameters: Ca2+ and Mg2+ ions. However, in the figure 3, Ca and Mg were used.
Comments on the Quality of English LanguageThe English could be improved to more clearly express the research.
Reviewer 2 Report
Comments and Suggestions for Authors
Second review of the manuscript entitled "Metallomics in placentae" (ID: ijms-3976490). I would like to thank the authors for providing a revised version of the manuscript. Here are a few more suggestions:
1. I didn't mention this in the previous review, but the title lacks a delineation of the nature of the study, as per the STROBE protocol.
2. Introduction. I believe that the text in lines 50-61 should be presented at the end of the introduction section. However, the text in lines 121-131 is unnecessary because, in my opinion, it is a fragment from a text in which the authors applied for funding for a research project, and therefore it has no application in a scientific study (this manuscript). Although I requested in the previous review that the repeated statement of the study's purpose be removed from the introduction, the authors still repeat the study's purpose at least twice, which is unnecessary. The introduction in scientific papers should be clear and concise, introducing the reader to the topic and presenting the gaps in the scientific evidence that the study fills. It's also important to remember that the manuscript is intended for professionals in the field, so it shouldn't be overly concise.
3. The authors still tend to repeat information presented once throughout the manuscript, e.g., lines 354-360: "A total of 33 samples of maternal placentae were obtained from pregnant women aged 21 to 40, who lived in Lower Silesia (Poland) and gave birth in the obstetric-gynecological ward of the First Department of Gynecology and Obstetrics of the Wroclaw Medical University in 2015-2016. The material was collected during natural delivery (n=2) or by caesarian section (n=31) in the obstetric-gynecological cesarean section (n=31) in the obstetric-gynecological ward of the First Department of Gynecology and Obstetrics of the Wroclaw Medical University." Is it necessary to repeat the research area twice? Of course, I have only cited a single example here; there are many more similar repetitions in the text that should be removed.
4. I believe that a p-value should be provided for Table 1. If the authors skillfully design the table, this will not affect the readability of the manuscript for the reader.
In summary, the manuscript, after the corrections introduced by the authors, better meets the rigor required for scientific papers. Despite the corrections, I believe the manuscript requires further improvement in accordance with the suggestions.
Round 3
Reviewer 1 Report
Comments and Suggestions for Authors
Thanks for author’s responses. However, English grammar of this manuscript STILL should be corrected by a Professional English proofreading and editing service. It is the third times to review this manuscript for this reviewer.
1. Line 32, correct ‘Mn²⁺ ions’ to ‘Mn²⁺’. Delete all superfluous ‘ion’. Please check and revise it throughout the manuscript. This reviewer has no enough time to point out one by one.
2. Lines 119 and 402, ‘magnesium (Mg2+ ion)’ or ‘Magnesium (Mg)’. Please check and revise others throughout the manuscript.
3. Line 129, correct ‘0.69μg/L’ to ‘0.69 μg/L’. Please check and revise it throughout the manuscript. This reviewer has no enough time to point out one by one.
4. Table 4, correct ‘29.5 ± 5,26’ to ‘29.5 ± 5.26’. Please check and revise it throughout the manuscript. This reviewer has no enough time to point out one by one.
5. Line 381, correct ‘HNO3’.
6. Conclusions section should be refined to one or two paragraphs.
7. The format of references should be based on MDPI style.
Author 1, A.B.; Author 2, C.D. Title of the article. Abbreviated Journal Name Year, Volume, page range.
Ref. 2, ‘J Clin Diagn Res’ vs Ref. 4, ‘J. Trace Elem. Med. Biol.’.
Ref. 4, ‘J. Trace Elem. Med. Biol.’ vs Ref. 24, ‘Biochimica et Biophysica Acta (BBA) - Molecular Cell Research’.
Comments on the Quality of English LanguageThe English could be improved to more clearly express the research.
Reviewer 2 Report
Comments and Suggestions for Authors
I thank the authors for incorporating the suggested corrections. I believe these were necessary to accurately and reliably present the study's findings. In its current form, I recommend the manuscript for publication in the journal.
Round 4
Reviewer 1 Report
Comments and Suggestions for Authors
Thanks for author’s responses. However, English grammar of this manuscript STILL should be corrected by a Professional English proofreading and editing service. It is the four times to review this manuscript for this reviewer.
1. Lines 27-28, correct ‘Ca–Mg, Fe–Zn, Mn–27 Cu’ to ‘Ca–Mg, Fe–Zn, and Mn–Cu’.
2. Line 29, correct ‘Hb, Cu–PLT, Cu–UA Pi)’. Please check and revise others throughout the manuscript.
3. Lines 55, 88, 196, ….., there are many ‘we’. As a scientific paper, writing should be done in the third person. Please check and revise it throughout the manuscript. This reviewer has no enough time to point out one by one.
4. Lines 128, 132, ‘Figure 1’ VS ‘Fig. 2’. Please check and revise it throughout the manuscript.
5. Line 132, Fig. 2, correct ‘μg/ g of tissue’.
6. Line 320, correct ‘2015-2016..’.
7. Table 4, correct ‘GDM 1’ and ‘GDM 2’ to ‘GDM1’ and ‘GDM2’.
8. The format of references should be based on MDPI style.
Author 1, A.B.; Author 2, C.D. Title of the article. Abbreviated Journal Name Year, Volume, page range.
Ref. 30, revise ‘Journal of Trace Elements in Medicine and Biology’.
Ref. 35, revise ‘Zahedan Journal of Research in Medical Sciences’. Please check and revise these throughout References section one by one.
Comments on the Quality of English LanguageEnglish grammar of this manuscript STILL should be corrected by a Professional English proofreading and editing service.
